# Quantitative Analysis of Differential Expression of HOX Genes in Multiple Cancers

**DOI:** 10.3390/cancers12061572

**Published:** 2020-06-14

**Authors:** Orit Adato, Yaron Orenstein, Juri Kopolovic, Tamar Juven-Gershon, Ron Unger

**Affiliations:** 1The Mina & Everard Goodman Faculty of Life Sciences, Bar-Ilan University, Ramat-Gan 5290002, Israel; orit.adato@biu.ac.il; 2School of Electrical and Computer Engineering, Ben-Gurion University of the Negev, Beer-Sheva 8410501, Israel; yaronore@bgu.ac.il; 3Department of Pathology, Hadassah Medical Center, Jerusalem 9112102, Israel; JuriK@hadassah.org.il

**Keywords:** *HOX* genes, cancer, transcription factors, differential expression analysis, patient survival

## Abstract

Transcription factors encoded by *Homeobox* (*HOX*) genes play numerous key functions during early embryonic development and differentiation. Multiple reports have shown that mis-regulation of *HOX* gene expression plays key roles in the development of cancers. Their expression levels in cancers tend to differ based on tissue and tumor type. Here, we performed a comprehensive analysis comparing *HOX* gene expression in different cancer types, obtained from The Cancer Genome Atlas (TCGA), with matched healthy tissues, obtained from Genotype-Tissue Expression (GTEx). We identified and quantified differential expression patterns that confirmed previously identified expression changes and highlighted new differential expression signatures. We discovered differential expression patterns that are in line with patient survival data. This comprehensive and quantitative analysis provides a global picture of *HOX* genes’ differential expression patterns in different cancer types.

## 1. Introduction

*Homeobox* (*HOX*) genes encode transcription factors that function as critical master regulators during embryogenesis in diverse processes, including apoptosis, receptor signaling, motility, and angiogenesis (reviewed in [1,2,3]). *HOX* genes have also been shown to function in adult stem cell differentiation [4,5,6].

The human *HOX* genes are organized in four genomic clusters marked by the letters A, B, C, and D, which are located on different chromosomes (7, 17, 12, and 2, respectively). The *HOX* genes of each cluster are numbered from 1 to 13. As some *HOX* genes in each locus were lost during evolution, there are a total of 39 *HOX* genes. *HOX* genes display spatial and temporal collinearity [7]. The *HOX* proteins are classified into 3 main groups—anterior (*HOX1-3*), central (*HOX4-8*), and posterior (*HOX9-13*), according to their expression along the anterior-posterior axis during development.

Multiple reports have demonstrated that mis-regulation of *HOX* genes expression plays key roles in the development of cancers. Only in the last five years have more than 50 reviews related to this subject been published; see, for example, References [4,8,9,10,11]. However, such studies were typically based on different technologies used in different labs and analyzed by different computational tools. Although some reviews summarized the findings of individual reports that analyzed the expression of distinct *HOX* genes in specific cancers (e.g., References [1,4]), a comprehensive quantitative comparison of *HOX* gene expression between cancer samples and samples obtained from healthy matched tissues is lacking.

Here, we present a thorough, standardized, uniform and systematic analysis performed using the most comprehensive data sources available to date: The Cancer Genome Atlas (TCGA) and the recently available Genotype-Tissue Expression (GTEx) resource [12]. Our standardized and systematic approach enabled us to perform rigorous comparisons between *HOX* gene expression data in different cancer types and matched healthy tissues, identify differential expression patterns that can confirm previously identified expression changes, and highlight new differential expression signatures, which we show are in line with patient survival data.

## 2. Results

### 2.1. Gene Expression Data Obtained from TCGA and GTEx Using Xena are Comparable

The TCGA database of cancer tissue expression levels does not include, by and large, samples of healthy tissues. Thus, our main strategy in this analysis is to compare expression data from cancer patients obtained from the TCGA database with healthy samples taken from the GTEx database. The developers of the UCSC Xena normalized the data to allow comparisons between these two data sources and indeed such a comparison was made in several studies (see, for example, References [13,14]). Nevertheless, in order to validate this approach, we first needed to establish that comparing data from these two sources is legitimate and does not introduce bias into the analysis.

For that purpose, we first analyzed gene expression either when both healthy and tumor data was obtained from TCGA, or when the tumor data was obtained from TCGA and the healthy data was obtained from GTEx. To this end, we used tissues that included at least 50 healthy samples in TCGA, namely breast invasive carcinoma (BRCA), liver hepatocellular carcinoma (LIHC), lung adenocarcinoma (LUAD), lung squamous cell carcinoma (LUSC), and thyroid carcinoma (THCA), which include 113, 50, 59, 50, and 59 TCGA healthy samples, respectively. To generate an empirical null distribution, we calculated the number of genes with differential expression between healthy and tumor breast tissue, for 1000 random sets of 39 genes (not specifically *HOX* genes). In each trial, we compared the lists of genes that were found to be differentially expressed in TCGA and GTEx. The calculation was done when both healthy and tumor data were obtained from TCGA and when the tumor data were taken from TCGA and healthy data were obtained from GTEx. The results demonstrate that the two data sources are comparable: the average difference in the number of differentially expressed genes is between 2 to 3 (Appendix A and Appendix A) and that *HOX* genes do not deviate differently from random sets of genes. When we specifically checked the number of *HOX* genes that were differentially expressed between tumor and healthy tissues using the healthy tissue from the two sources we found that for BRCA, LIHC, LUAD, LUSC & THCA the difference was 4, 4, 3, 3, and 7 genes, respectively (with empirical *p*-value 0.321, 0.221, 0.711, 0.49, and 0.004, respectively) (Appendix A and Appendix A). We assume that the large deviation for thyroid carcinoma stems from different relative abundance of the tumor subtypes: (papillary thyroid cancer, follicular thyroid cancer, anaplastic thyroid cancer, medullary thyroid cancer) in the two databases, but we did not have clinical documentation to analyzed this issue further.

These observations suggest that the comparison of gene expression data of tumor samples obtained from TCGA to gene expression data of healthy samples obtained from GTEx using Xena is valid.

### 2.2. Differential Expression of HOX Gene Clusters Can Discriminate Between Tumor Samples and Healthy Samples

Following the assertion that GTEx data can be used as a source for healthy samples, we compared *HOX* gene clusters in tumor and healthy samples. Fourteen cancer types, for which there were at least 100 tumor samples (in TCGA) and at least 100 healthy samples (in GTEx), were included in this analysis (Table 1). For each of these 14 cancer types, namely cancers associated with the adrenal gland, brain, breast, colon, esophagus, blood cells (leukemia), liver, lung, pancreas, prostate, stomach, and thyroid, the normalized expression levels of 39 *HOX* genes were obtained. A hierarchical clustered heatmap was created based on the average Euclidean distance between healthy samples compared with the average Euclidean distance of tumor samples. As an example, we present 3 such heatmaps (Figure 1, Figure 2 and Figure 3). The other heatmaps can be found in the Appendix A. The heatmaps demonstrate that, in all cancer types, differential expression of *HOX* gene clusters can separate tumor samples from healthy samples and that, for some of the *HOX* genes, the expression changes are higher than that of others (Figure 1, Figure 2 and Figure 3 and Appendix A).

### 2.3. Differential Expression of HOX Genes in 14 Different Cancer Types

To determine which *HOX* genes significantly change their expression levels in tumor tissues compared to healthy tissues, we performed the statistical Wilcoxon rank-sum test for every *HOX* gene and for every cancer type. A 2-fold differential expression cut-off was used for binary classification of differently expressed genes, i.e., *HOX* genes with expression in cancer samples statistically significant as higher than twice the expression in healthy samples, or lower than half the expression in healthy samples, were considered as “differentially expressed genes”.

In all 14 analyzed cancer types, there were multiple *HOX* genes with significant (*p* < 0.05 Bonferroni adjusted for *n* = 39) altered expression between healthy and tumor tissues (Figure 4).

Note that, in brain tumors, glioblastoma multiforme (GBM), a large number of *HOX* genes are differentially expressed between tumor and healthy samples (36 *HOX* genes). In the GTEx project design details, it was indicated that the majority of the brain and brainstem were left unfixed and shipped overnight on wet ice to a brain bank [15]. We ruled out the possibility that low expression of *HOX* genes in healthy samples of the brain is an artifact of the way that these samples were treated by showing that the expression level of other sets of random 39 genes were in the normal range. Notably, there were 6 *HOX* genes, namely *HOXA2*, *HOXA4*, *HOXB2*, *HOXB3*, *HOXB4*, and *HOXC4*, which changed expression only in brain cancer (either in GBM, brain lower grade glioma (LGG), or in both).

In addition to GBM, there were 5 cancer types for which a third or more of the *HOX* genes displayed altered gene expression, namely LGG, esophageal carcinoma (ESCA), LUSC, pancreatic adenocarcinoma (PAAD), and stomach adenocarcinoma (STAD). Intriguingly, 4 out of these 5 cancer type tissues are derived from endodermal organs that are located in the foregut.

As mentioned above, *HOX* genes are organized in 4 genomic clusters (*HOXA*, *HOXB*, *HOXC*, *HOXD*), and genes within each cluster can be grouped based on their expression in different developmental regions (anterior, central, or posterior). Appendix A provides the type of expression change (upregulation/downregulation/no change) for each analyzed cancer type, with *HOX* genes organized into clusters and groups. Interestingly, the majority of *HOX* genes (94 out of 160) that change expression belong to the posterior *HOX* group (*HOX9-13*) (Appendix A). Notably, calculation and comparison of the mean expression level of 1000 randomly selected 16 *HOX* genes to the mean expression levels of the 16 posterior *HOX* revealed that many posterior *HOX* genes are expressed at low levels in multiple healthy tissues (empirical *p*-value = 0.013) (Appendix A), in line with PCR analysis of *HOX* gene expression in normal thyroid gland, performed by Takahashi et al. [16]. The “anterior *HOX* group” is the group with the least number of *HOX* genes with altered gene expression in cancer tissues—18 genes out of 160 differentially expressed genes (these 18 genes account for ~16% of the total number of anterior genes in 14 cancer). Most of the *HOX* genes (150 out of 160) with altered expression in the analyzed cancer tissues were upregulated. However, in colon adenocarcinoma (COAD), acute myeloid leukemia (LAML), pheochromocytoma and paraganglioma (PCPG), prostate adenocarcinoma (PRAD), and THCA, there were a total of 10 *HOX* genes that were downregulated (Figure 4, Appendix A). *HOXC* is the *HOX* cluster with the highest number of *HOX* genes that change expression (48 genes). Most of the *HOXC* cluster genes display relatively low expression levels in the analyzed healthy tissues (Appendix A). In this analysis, *HOXA11* and *HOXA13* were identified as differentially expressed in the highest number of cancer types (10 out of 14).

We next focused on the genes that undergo the most significant changes (Table 2). We highlighted genes with at least a 3-fold expression change between healthy and tumor tissues, and with normalized gene expression levels in healthy tissues of at least 1 unit (RNA-seq by Expectation Maximization (RSEM) normalized count, log2(x + 1) transformed). This requirement was implemented in order to identify genes whose expression levels changed significantly, both in terms of fold change and in terms of absolute levels. *HOX* genes that met these criteria were found in 9 out of 14 analyzed cancer types: BRCA, ESCA, GBM, LAML, LGG, LIHC, LUSC, PAAD, and STAD. Seventeen out of 28 cases that met the above criteria for differentially expressed genes belong to the ‘posterior *HOX* group’, 10 belong to the ‘central *HOX* group’, and one gene belongs to the ‘anterior *HOX* group’. The highest expression fold change was observed for *HOXC10* in STAD. Notably, there were 3 other *HOX* genes with a fold change greater than 4 in STAD: *HOXA10*, *HOXB9*, and *HOXC8*.

We noted that some differentially expressed *HOX* genes are expressed at very low levels in healthy tissues (less than 1 unit of normalized expression values; Xena) and have high expression fold-change. To identify the genes whose expression is significantly altered and their absolute expression levels in tumor tissues is not very low, we applied selection criteria to highlight the genes whose expression levels in tumor tissues were higher than 1 unit of normalized expression values and had at least a 3-fold expression change between healthy and tumor tissues (Appendix A). The cancer types with the highest number of *HOX* gene that met the above criteria were in GBM (25 genes), while the highest expression changes were observed in STAD (Appendix A). Notably, most of the *HOX* genes that met the above criteria belong to the posterior *HOX* group (38 out of 57 cases), and among them, *HOXA13* stands-out as its expression changed in most cancer types (6 out of 14) (Appendix A).

### 2.4. Kaplan-Meier Survival Analysis

The differentially expressed genes that most significantly changed (Table 2, Appendix A) and also demonstrate correlation with patients’ survival may potentially serve as prognostic markers. To identify candidate genes that can serve as potential prognostic markers, a Kaplan-Meier (KM) survival analysis was performed for every differentially expressed gene in every cancer type (Appendix A). Table 3 lists the genes for which the KM analysis results were statistically significant (*p*-value < 0.05 Bonferroni adjusted for *n* = 39).

In 14 out of the 160 cases in which a *HOX* gene was differentially expressed in one of the 14 analyzed cancer types, there was a significant correlation between increased expression levels and poor survival (*p*-value < 0.05 Bonferroni adjusted for *n* = 39) (Appendix A).

We next examined whether pairs of *HOX* genes could be used together as stronger prognosis markers. To this end, a KM survival analysis was performed for every cancer type, for every combination of *HOX* pair where both *HOX* genes were differentially expressed. The pairs that demonstrated significant results (*p*-value < 0.05 Bonferroni adjusted for *n* = 741), were compared with the KM survival analysis performed for each one of the paired genes separately (Appendix A). Such pairs were found in 2 cancer types, LGG and GBM (20 and 4 pairs, respectively). For example, LGG *HOX* pair *HOXA1* and *HOXD10* demonstrates a more statistically significant KM curve than the KM survival curve performed for each one of the paired genes separately (Figure 5). Details of significant pairs are included in Appendix A. Importantly, each of these gene pairs could potentially serve as a prognostic marker.

### 2.5. HOX Gene Pairs with Correlated Differential Expression

Genes that have combined effects or act in similar pathways may demonstrate an expression correlation. The expression analysis results included several cases in which in the same cancer type, there were a few genes that demonstrated a similar fold change between tumor and healthy tissue (Figure 4, Table 2). For example, in ESCA, *HOXC13*, and *HOXD11* demonstrated~5-fold expression change, or in PAAD, *HOXA10*, *HOXB8*, *HOXC8*, and *HOXC9* demonstrated~3-fold expression change (Table 2). Based on these findings, and following the results of the KM analysis, we next examined whether there are any *HOX* genes pairs that show correlated expression patterns within the same sample in both tumor and healthy tissues. To this end, a Pearson correlation was calculated for all possible 741 *HOX* gene pairs for each analyzed cancer type. The analysis was performed for gene pairs for which both tumor and healthy cohorts demonstrated normal distribution. These genes were identified by Kolmogorov–Smirnov normality test. The gene pairs that change expression between healthy and tumor samples and show moderate or strong correlation (i.e., either *r* > 0.4 or *r* < −0.4 [17]) in both tissue types were considered correlated. The resulting gene pairs were observed in 6 cancer types: ESCA, GBM, LAML, LIHC, LUSC, and STAD (Table 4, correlation graphs are included in Appendix A). In 14 (out of 19) gene pairs, both pair members belong to the same cluster (Table 4, Appendix A). Notably, all resulting mixed clusters pairs include one gene from cluster A and one gene from another cluster (Table 4, Appendix A). Examination of the pair composition from the *HOX* group perspective indicates that only 9 (out of 19) pairs belong to the same *HOX* group (Appendix A).

The expression correlation observed between *HOXB5*, *HOXB6*, and *HOXB7* in ESCA, is in line with a previous report that showed that expression changes in the three “midcluster” *HOXB* genes, namely *HOXB5*, *HOXB6*, and *HOXB7*, can trigger changes in the transcriptional program of adult esophageal cells, with implications to the early stages of esophageal carcinogenesis [18]. It is possible that since these genes are neighboring genes they are regulated by the same factors and therefore manifest expression correlations.

Similarly, we wanted to check whether there are *HOX* gene pairs that show moderate or strong expression correlation in healthy tissue and lose it in tumor tissue, or vice versa. Out of 69 *HOX* gene pairs that show weak expression correlation (i.e., either *r* < 0.4 or 0 > *r* > −0.4) in healthy and moderate or strong correlation in tumor tissue, there were 21 pairs that showed strong correlation in tumor tissue (i.e., either *r* > 0.7 or *r* < −0.7). Moreover, 6 of these gene pairs were *HOXA* pairs observed in LAML with Pearson correlation coefficient *r* > = 0.89 (Appendix A). Interestingly, there were only 7 *HOX* gene pairs that showed moderate expression correlation (0.4 < *r* < 0.7) in healthy tissue and did not show expression correlation in tumor tissue (either *r* < 0.4 or 0 > *r* > −0.4).

### 2.6. Comparison of HOX Genes Expression with Expression of House Keeping Genes

To validate that the differential expression pattern of *HOX* genes in tumor tissue is not a generic behavior of genes in tumor tissue, we examined the expression pattern of housekeeping (HK) genes in healthy and tumor tissue and compared it with *HOX* genes’ expression pattern. For every cancer type, a differential expression analysis was performed for a list of 3804 human HK genes compiled by Eisenberg et al. [19]. The vast majority of the HK genes are not differentially expressed between tumor and healthy samples. For every cancer type there were very few (10 or less out of 3804) HK genes that differentially expressed (Appendix A). Moreover, in all analyzed cancer types, the expression levels of HK genes were higher (2-fold or more) than the expression level of *HOX* genes (empirical *p*-value = 0).

### 2.7. Comparison of HOX Genes Expression with Expression of Transcription Factors Enocoding Genes in Brain Healthy and Tumor Tissue

As cells from healthy brain tissue present virtually no proliferation, expression of a huge number of genes is activated in glioblastomas. Therefore, we performed a differential expression analysis between brain healthy tissue and GBM tumor tissue for other transcription factors (TF) and compared their expression pattern with *HOX* genes expression pattern. The analysis was performed on a list of human transcription factors compiled by Lambert SA et al. [20]. The *HOX* genes are a subgroup of the Homeodomain TF family. In GBM, most of the *HOX* genes (36 out of 39) are differentially expressed between tumor and healthy tissues, whereas a smaller fraction of the non-*HOX* Homeodomain genes (43 out of 187) is differentially expressed (Appendix A). Remarkably, in other TF gene families, such as bZIP and Nuclear Receptors, most of the genes are expressed in both healthy and tumor tissues and demonstrate little expression changes (Appendix A).

### 2.8. Comparison of HOX Genes with Known Prognostic Markers

Many of the cancer-related prognostic markers used in medical clinics are not based on changes in gene expression level. In order to compare *HOX* gene expression patterns in cancer tissue with known cancer markers, we searched for prognostic markers that are known to undergo genetic alterations and protein level changes. We have compiled a list of 29 previously suggested prognostic markers genes for the 8 analyzed cancer types in which 11 or more (about a third) of the *HOX* genes were found to be differentially expressed (Appendix A) [21,22,23,24,25,26,27,28,29,30,31,32]. For each marker, we examined whether it is differentially expressed between healthy and tumor tissue and performed a Kaplan-Meier (KM) survival analysis. The analysis demonstrated that only 6 of the previously suggested prognostic markers included in the list, namely *ABCC3* in GBM, *CDX2, KIT*, and *WT1* in LAML and *CEACAM5* in LIHC and STAD, are differentially expressed between tumor and healthy tissue, and 3 of them (*ABCC3*, *KIT*, and *WT1*) demonstrate a significant correlation between increased expression levels and poor survival (*p*-value < 0.05) (Appendix A).

### 2.9. HOX Expression in BRCA HER2-Positive, BRCA HER2-Negative, and BRCA Triple-Negative Samples

Analysis of *HOX* genes expression in ‘BRCA Human Epidermal Growth Factor Receptor 2 (HER2)-positive’ tumor samples and ‘BRCA HER2-negative’ tumor samples, revealed significant differences in the expression levels of 8 *HOX* genes (Appendix A). Interestingly, the expression level of *HOXB13* in BRCA HER2-positive samples is~4-fold higher than its expression in HER2-negative samples.

A similar analysis performed for *HOX* genes expression in ‘BRCA Triple-negative’ tumor samples and ‘BRCA non-Triple-negative’ tumor samples revealed significant differences in the expression levels of 21 genes, yet all differences were less than 2-fold (Appendix A).

## 3. Discussion

This work includes a comprehensive quantitative analysis of 14 cancer types providing an overview picture of *HOX* gene expression pattern in these cancers, using a consistent source of data (TCGA for tumor samples and GTEx for healthy samples). It includes at least 100 samples of every cancer type and highlights the expression differences between tumor and healthy samples.

The results show that differential expression of *HOX* gene clusters can discriminate between tumor samples and healthy samples (Appendix A). In all analyzed cancer types, there were several *HOX* genes that demonstrated expression change of 2-fold or higher between healthy and tumor tissue. These results support the hypothesis raised by Bhatlekar et al. [1] that there may be *HOX*-related regulatory networks that become dysregulated during cancer development.

By and large, our findings agree with previous reports about the roles specific *HOX* genes play in cancers. However, we highlighted the involvement of *HOX* genes and gene pairs that were not previously implicated in cancer. There are also cases where our findings do not support previous reports, for example the involvement of several *HOX* genes in PRAD mentioned in [1], which we did not observe. It is clear that different studies (including ours) may have their own pitfalls and limitations, such as those discussed in a recently published study focusing on *HOX* and three amino acids loop extension (TALE) homeodomain transcription factors expression in cancer [33] that may stem from the amount of samples included in the analyses, the heterogeneity of the samples and the analyses methodology. However, our systematic and comprehensive analysis, implementing the same analysis method on all cancer types, provides a better global overview of the role of *HOX* gene expression in these cancers.

While the observed expression changes in *HOX* genes were both upregulation and downregulation, in the majority of cancer types the significant expression changes were towards upregulated expression. The analysis showed that in GBM, most of the *HOX* genes were differentially expressed. This can be attributed to the fact that *HOX* genes are hardly expressed in normal brain tissue (Appendix A). Thus, we suggest that in tumor tissues there is a change in a “master regulator pathway” that controls *HOX* genes expression and limits their expression in healthy tissues. The *HOX* genes belong to homeodomain TF group. Notably, in the comparison of their expression pattern to the expression pattern of other TF groups in GBM tumor tissue and healthy tissue, it was observed that unlike the *HOX* genes, most of the TF belonging to other TF groups, such as bZIP and Nuclear Receptors, are expressed in both tumor and healthy tissue to similar levels. We also observed that the expression level of *HOX* genes is higher in GBM samples than in LGG samples (Figure 3 and Figure 4, Appendix A). This observation could imply that in brain tissue, higher expression levels of *HOX* genes correlate with a more aggressive malignant tumor. Previous studies have shown that upregulation of different *HOX* genes can be used as gene signatures in glioblastoma [34,35,36]. However, it would be interesting to perform further analysis on samples from different Glioma stages, as additional supportive evidence could indicate that the expression levels of *HOX* genes may be used as a marker to determine the stage and progression of brain cancer.

Our analysis demonstrates that *HOX* genes are cooperatively involved in 6 different types of cancers. We showed which pairs of genes are co-up or -down regulated. The fact that in most of *HOX* gene pairs that demonstrate expression correlation both pair members belong to same cluster supports the hypothesis that they share a similar regulator. Yet, in order to better understand how *HOX* genes that are co-expressed influence tumor development, further mechanistic studies addressing their regulation are required.

*HOX* genes that undergo the most significant changes and also demonstrate significant correlations between gene expression and patients’ poor survival can be potential prognostic biomarkers. The genes marked in * bold in Table 2 and in Appendix A are those that met these 2 criteria. These genes were already suggested for consideration as prognostic biomarkers in previous studies (selected references are listed in Appendix A [34,37,38,39,40,41,42,43,44,45,46,47,48,49,50,51,52,53,54,55,56,57,58]).

Interestingly, the comparison of the differential expression pattern of *HOX* genes between tumor and healthy tissue with some of the known prognostic markers that are also known to undergo genetic alterations and protein expression changes, showed that for most of these prognostic markers the expression change is less than 2-fold (the cutoff used in our analysis), and more than half of these genes do not demonstrate significant correlation between expression levels in tumor tissue and patients’ survival.

The importance of the cooperative effect of *HOX* genes is also demonstrated by our finding that specific pairs of *HOX* genes, whose expression levels are both high or low, can serve as better markers for survival than what can be predicted from each gene separately (Appendix A).

Most of the *HOX* genes that change expression in tumor tissues belong to the posterior *HOX* group. Since in most analyzed healthy tissues, the expression levels of posterior *HOX* genes are very low, one can speculate that posterior *HOX* genes have a bigger contribution to the development of cancer than anterior *HOX* genes.

Finally, apart from GBM, 4 of the cancer types for which a third or more of the *HOX* genes displayed altered gene expression (namely: ESCA, LUSC, PAAD, and STAD), originated from endodermally derived organs that are located in the foregut. Interestingly, these cancer types are known to be aggressive with low survival rates [50].

## 4. Materials and Methods

### 4.1. Data Downloads

The source of expression data files used in this analysis originated from UCSC Xena [59]. The analyzed data included gene expression data of tumor samples from the cancer genome atlas (TCGA) and gene expression data of healthy samples from Genotype-Tissue Expression (GTEx) downloaded from UCSC Xena data hubs [60]. Specifically, the data used in the analysis of differential *HOX* gene expression in different cancer types were obtained from the ‘UCSC Toil RNAseq Recompute’ data hub. Under this data hub, from “TCGA TARGET GTEx” cohort, the RSEM_norm_count file was downloaded from the “gene expression RNAseq” section, and the TCGA_GTEX_main_categories file was downloaded from the “phenotype” section. For the differential expression analysis of *HOX* genes, the samples from TCGA and GTEx were selected.

A separate analysis was performed in order to compare the expression of randomly selected genes in breast cancer patients. The expression data for this analysis included breast cancer samples from both tumor and healthy tissues obtained from the TCGA hub. The IlluminaHiSeq file was downloaded from the TCGA data hub, under the “TCGA Breast Cancer (BRCA)” cohort, and under the “gene expression RNAseq” section.

The list of housekeeping genes was downloaded from Reference [19].

The list of human transcription factor genes was downloaded from [20].

Clinical data for BRCA tumor samples was downloaded from the ‘Phenotypes TCGA hub’ located under UCSC Xena, ‘TCGA Breast Cancer (BRCA)’ cohort.

### 4.2. Association of Samples to Cancer Types

Each sample in the downloaded data was identified by a unique ID and categorized by tissue site and primary disease type in the case of tumor samples, or tissue site and tissue type in the case of healthy samples. The TCGA cohort included tumor samples belonging to 33 different cancer types and very few samples taken from healthy tissue. In this analysis only TCGA tumor samples were used, i.e., the few TCGA normal samples were removed. The GTEx cohort included samples from 54 different types of healthy tissues. The tissue categories in the TCGA cohort were slightly different from the tissue categories in the GTEx cohort. In order to compare gene expression from tumor samples with the expression in the corresponding healthy tissues, a joint category table was built. The joint category table matched the TCGA tumor tissue categories with their corresponding GTEx categories of healthy tissues (Appendix A) and assigned them to a joint category. Every category in the table represents a cancer type. Based on this table, the samples were divided to 18 files. Importantly, the analysis was only performed on cancer types for which there were at least 100 tumor samples and at least 100 healthy samples. Fourteen out of the 18 files of different cancer types met these criteria. The expression data included in this analysis was associated with tissues from the following organs: adrenal gland, brain, breast, colon, esophagus, blood cells (leukemia), liver, lung, pancreas, prostate, stomach, and thyroid. Table 1 lists the organs, the total number of TCGA samples, and the total number of GTEx samples included in the analysis.

### 4.3. Heatmaps and Data Processing

Heatmaps and dendograms were constructed with R statistics version 3.4.3, using heatmap3 package standard settings [61]. Average linkage two ways hierarchical clustering with Euclidean distance were used for the construction of heatmaps. Other data processing was performed using scripts written in python 2.7 [62]. Code for the analysis of this study is available at https://github.com/OritAdato/HoxAnalysis.

### 4.4. Comparison of Expression of Randomly Selected Genes Between Cancer Tissue and Healthy Tissue

This analysis was performed on normalized RNA-seq expression data (RSEM normalized count, log2(x + 1) transformed) of tumor and healthy tissue downloaded from two different UCSC Xena data hubs: TCGA hub and “Toil RNAseq Recompute” hub. In the differential expression analysis of the tissue samples downloaded from the TCGA hub, both tumor and healthy samples were from TCGA, while in the differential expression analysis of the tissue samples downloaded from “Toil RNAseq Recompute” hub, the tumor samples were from TCGA, and the healthy samples were from GTEx [63]. This analysis was performed for five tumors types for which at least 50 healthy samples were available in TCGA (BRCA, LIHC, LUAD, LUSC, and THCA).

To generate a null distribution for *HOX* gene sets, we randomly picked sets of 39 genes. For 1000 randomly selected sets of 39 genes, we compared the changes in gene expression levels between tumor and healthy samples. Every comparison using a random set of 39 genes was performed twice: once for comparing a cohort of tumor samples from TCGA with healthy samples from TCGA, and a second time for comparing the same TCGA tumor samples with healthy samples from matching tissues in GTEx. For every gene expression comparison, Wilcoxon rank-sum test was run to check whether expression of healthy samples differs from expression of tumor samples. A two-fold cutoff was used to define differential expression. The list of differentially expressed genes from the comparison between tumor samples and TCGA healthy samples (*p*-value < 0.05 (Bonferroni adjusted to *n* = 39) and expression change >2-fold), was compared with the list of the differentially expressed genes resulting from the comparison between tumor samples and GTEx healthy samples. We compared the number of genes that were unique to either gene list (i.e., genes that were not included in the intersection between the two lists).

### 4.5. Calculating Euclidean Distance Between Healthy and Tumor Samples

Using R statistics version 3.4.3, the average intra Euclidean distance within the healthy samples was calculated and compared with the average inter Euclidean distance between healthy samples and tumor samples. The comparisons were performed on tumor and healthy samples using Wilcoxon rank-sum test.

### 4.6. Verifying the Observation that Posterior HOX Genes are Expressed at Low Levels in Multiple Healthy Tissues

For every *HOX* gene, across all 14 analyzed healthy tissues, the mean expression level was calculated. As there are a total of 16 posterior *HOX* genes (namely, *HOXA9*, *HOXA10*, *HOXA11*, *HOXA13*, *HOXB9*, *HOXB13*, *HOXC9*, *HOXC10*, *HOXC11*, *HOXC12*, *HOXC13*, *HOXD9*, *HOXD10*, *HOXD11*, *HOXD12*, and *HOXD13*), as an empirical null distribution, the mean expression level of 1000 randomly selected 16 *HOX* genes was calculated and compared to the mean expression level of the 16 posterior *HOX* genes.

### 4.7. Kaplan-Meier Survival Analysis Based on Expression of HOX Genes

To check whether *HOX* genes that change expression between tumor and healthy samples demonstrate a correlation with poor survival, a Kaplan-Meier (KM) survival plot was created for every differentially expressed gene in every cancer type indicated in Table 1. The KM plots were generated using UCSC Xena browser based on TCGA tumor samples’ survival data included in ‘TCGA TARGET GTEX’ study. The reported p-values were corrected using Bonferroni correction for multiple-hypothesis test. For every analyzed gene, tumor samples were first divided to 2 groups based on the median expression. If two group analysis result was not significant, then an additional analysis was performed on 3 groups of samples divides based on expression: the upper third, middle third, and lower third. Lastly, if 3 group analysis was not significant, then a KM analysis on upper quartile and lower quartile was performed.

### 4.8. KM Survival Analysis Based on Expression of HOX Gene Pairs

To search for pairs of *HOX* genes that can predict prognosis of cancer patients better than each *HOX* separately, a KM survival analysis was performed for all combinations of *HOX* gene pairs, in every cancer type. For every gene in a pair, the samples were divided to two groups: the group of samples in which the gene expression is higher than the median and a group in which the gene expression is lower than the median. The KM analysis of each gene pair compared 2 groups of samples: the group created by intersection of the 2 groups of samples with expression higher than the median expression of every gene (marked in red in KM plots and in the risk tables below the plots; group size denoted as N1), and the group created by intersection between the samples with expression lower than the median expression of every gene (marked in blue in KM plots and in the risk tables below the plots; group size denoted as N2). In addition, a KM survival analysis was performed for each gene separately. For this single gene analysis, we took the N1 samples with the highest expression and the N2 samples with the lowest expression. The survival analysis of the pair of genes was then compared with the survival analysis of every one of the genes separately. We focused on gene pairs with KM-based *p*-value < 0.05 (Bonferroni-corrected for multiple gene pairs) and a *p*-value smaller than the *p*-value of the survival analysis of each of the two genes separately.

### 4.9. Comparison of Expression of Randomly Selected Sets of 39 House Keeping Genes and HOX Genes

For every cancer type, 1000 sets of 39 HK genes were randomly selected. The GTEx median expression level and the TCGA median expression level was calculated for every set and compared with the GTEx median expression level and TCGA median expression of the 39 *HOX* genes, respectively.

### 4.10. Analysis of HOX Genes Expression in BRCA HER2-Positive, BRCA HER2-Negative and BRCA Triple-Negative Samples

The clinical data information regarding HER2 status, Progesterone status, and Estrogen status was correlated to every BRCA tumor sample based on sample ID. Tumor samples with status NA or status different than ‘negative’ or ‘positive’ were excluded from the analysis. For every *HOX* gene, the statistical Wilcoxon rank-sum test was performed to compare the expression between BRCA HER2-negative tumor samples and BRCA HER2-positive tumor samples.

Samples that had a negative status indication for HER2, Progesterone, and Estrogen were marked as triple-negative. The statistical Wilcoxon rank-sum test was performed to compare expression between triple-negative BRCA tumor samples and non-triple-negative BRCA tumor samples.

## 5. Conclusions

This comprehensive and quantitative analysis of *HOX* gene expression in healthy human tissues and 14 different cancer types, provides a global picture of the *HOX* genes expression patterns in the analyzed cancer tissues. Importantly, the quantitative characterization of specific differentially expressed *HOX* genes highlights specific *HOX* genes and *HOX* gene pairs, which can serve as novel biomarkers to discriminate between healthy and tumor tissues. Furthermore, the identification of specific *HOX* genes that are differentially co-expressed in distinct cancer types can serve as a basis for future mechanistic studies, which would contribute to the understanding of their regulation in health and disease.

## Figures and Tables

**Figure 1 cancers-12-01572-f001:**
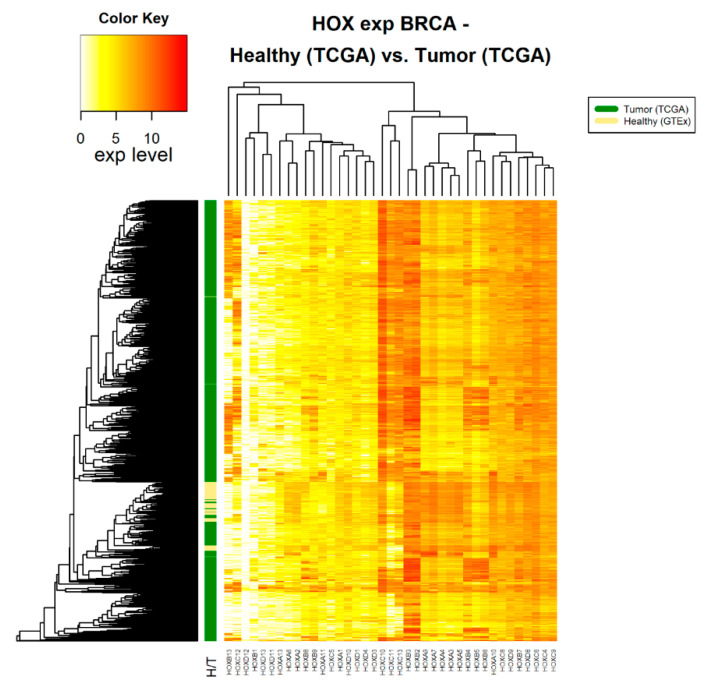
*HOX* gene expression in healthy and tumor breast tissue samples. The source of expression data of both healthy and tumor samples is TCGA.

**Figure 2 cancers-12-01572-f002:**
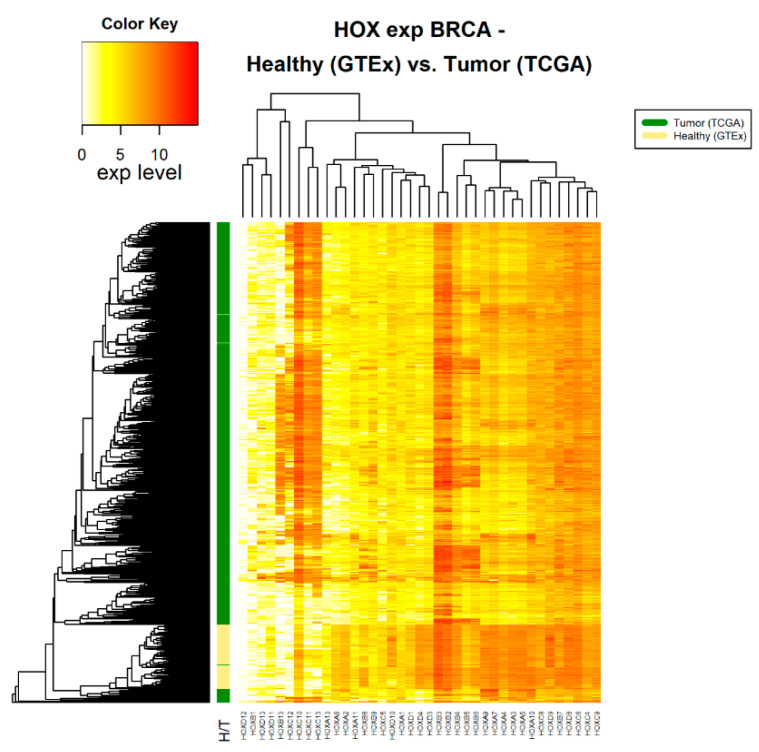
*HOX* gene expression in tumor and healthy breast tissue samples. The source of expression data of healthy and tumor samples is GTEx and TCGA, respectively.

**Figure 3 cancers-12-01572-f003:**
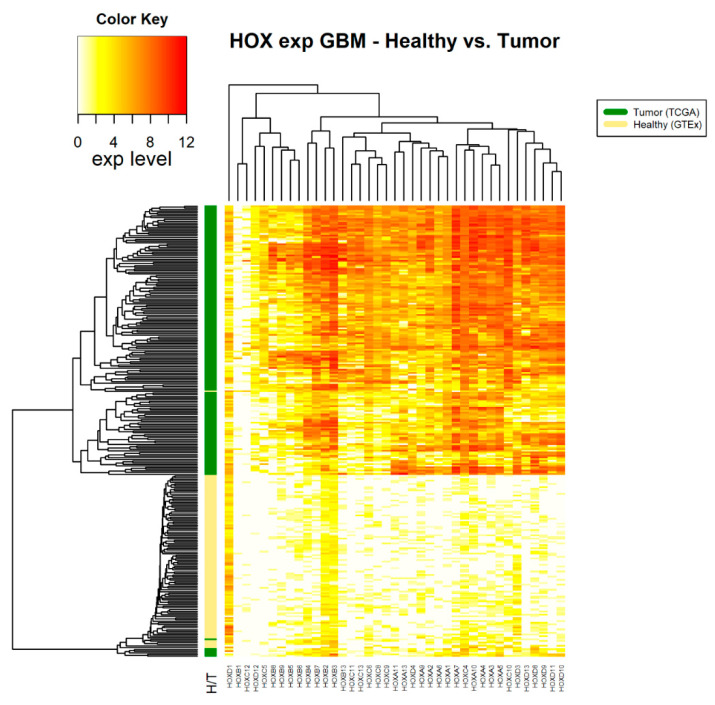
*HOX* gene expression in samples of healthy brain tissue and glioblastoma multiforme tumor tissue. The source of expression data of healthy and tumor samples is GTEX and TCGA, respectively.

**Figure 4 cancers-12-01572-f004:**
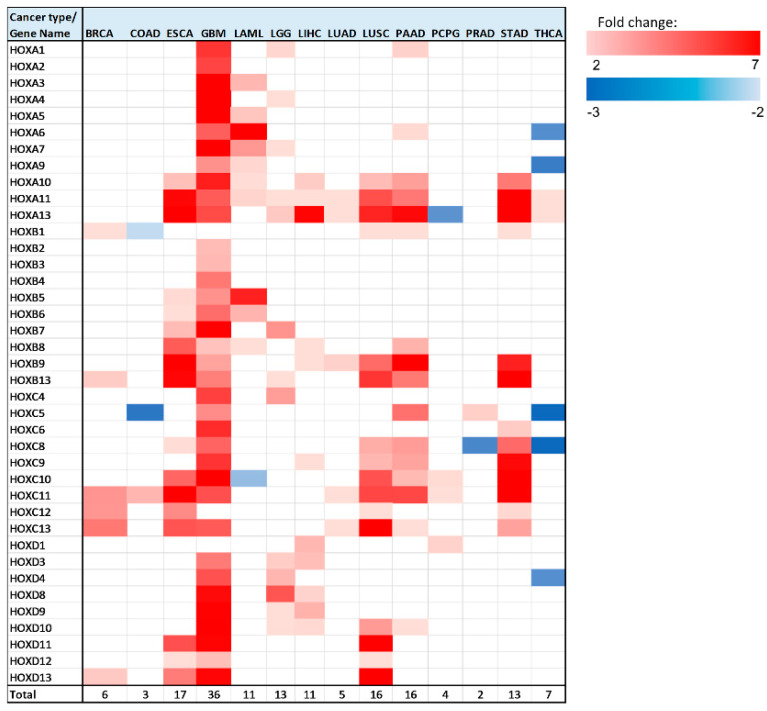
Heatmap of differentially expressed *HOX* genes in different cancer types. Gene column includes list of all 39 *HOX* genes tested in this analysis. The rest of the columns represent the analyzed cancer types. Upregulated genes are marked in red color gradient (light red represents lower fold change), downregulated genes are marked in blue color gradient (light blue represents lower fold change).

**Figure 5 cancers-12-01572-f005:**
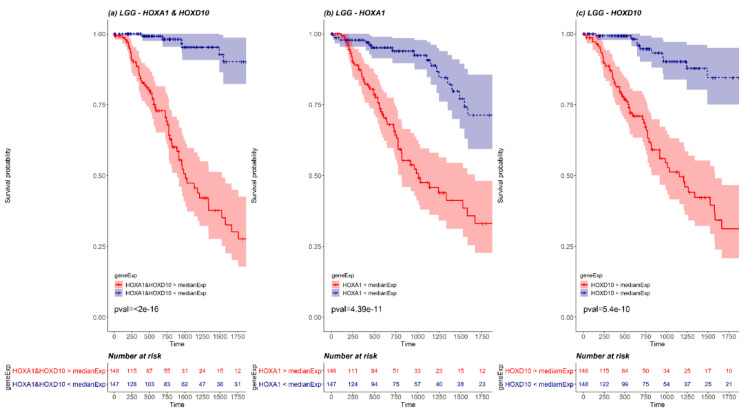
A KM plot of *HOXA1* and *HOXD10* pair (**a**) versus separate KM plots for each of *HOXA1* (**b**) and *HOXD10* (**c**) in LGG. The red line represents the survival of patients with expression higher than median expression, and the blue line represents the survival of patients with expression lower than median expression. The pink and the light blue background of the lines mark the confidence interval. Out of 527 LGG patients, there were 148 patients with both *HOXA1* and *HOXD10* expressed above LGG median expression (marked in red in the risk table below graph) and 147 patients with both *HOXA1* and *HOXD10* expressed below LGG median expression (marked in blue in the risk table below graph).

**Table 1 cancers-12-01572-t001:** The number of Genotype-Tissue Expression (GTEx) and The Cancer Genome Atlas (TCGA) samples used in the study ^1^.

Organ/Cancer Tissue Type	GTEX	TCGA	Total
Adrenal_Gland/PCPG	128	185	313
Brain/LGG	105	527	632
/GBM		173	173
Breast/BRCA	179	1253	1432
Colon/COAD	308	455	763
Esophagus/ESCA	273	195	468
Leukemia/LAML	337	173	510
Liver/LIHC	110	422	532
Lung/LUAD	288	598	886
/LUSC		551	551
Pancreas/PAAD	167	183	350
Prostate/PRAD	100	556	656
Stomach/STAD	175	451	626
Thyroid/THCA	280	571	851
Total	2450	6293	8743

^1^ The number of healthy samples is listed under the GTEx column and the number of tumor samples is listed under TCGA column. Acronyms of cancer tissue types: BRCA = breast invasive carcinoma, COAD = colon adenocarcinoma, ESCA = esophageal carcinoma, GBM = glioblastoma multiforme, LAML = acute myeloid leukemia, LGG = brain lower grade glioma, LIHC = liver hepatocellular carcinoma, LUAD = lung adenocarcinoma, LUSC = lung squamous cell carcinoma, PAAD = pancreatic adenocarcinoma, PCPG = pheochromocytoma and paraganglioma, PRAD = prostate adenocarcinoma, STAD = stomach adenocarcinoma, and THCA = thyroid carcinoma.

**Table 2 cancers-12-01572-t002:** *HOX* genes with pronounced expression change between healthy and tumor tissues ^1^.

Gene Name	Cancer Type								
	BRCA	ESCA	GBM	LAML	LGG	LIHC	LUSC	PAAD	STAD
*HOXA5*			7.0						
*HOXA7*				**3.6 ***					
*HOXA9*			3.7						
*HOXA10*			6.2					3.4	4.3
*HOXB4*			4.3						
*HOXB8*								3.0	
*HOXB9*							4.6		6.2
*HOXB13*		6.8							
*HOXC4*			5.5		**3.5 ***				
*HOXC6*			5.9						
*HOXC8*							3.2	3.5	4.6
*HOXC9*								3.4	
*HOXC10*		4.7					5.1		7.5
*HOXC11*	3.7								
*HOXC13*	4.3	5.1							
*HOXD3*			4.3						
*HOXD9*						3.0			
*HOXD10*							3.6		
*HOXD11*		5.2							

^1^ The expression fold change of *HOX* genes with at least a 3-fold expression change between healthy and tumor tissues and with gene expression levels of at least 1 unit (RSEM normalized count, log2(x + 1) transformed) in healthy tissues. The cells marked * bold highlight the genes for which the Kaplan-Meier (KM) graph showed a significant correlation between gene expression and patients’ survival (analysis detailed below).

**Table 3 cancers-12-01572-t003:** *HOX* genes with statistically significant KM curve ^1^.

Organ/Cancer Type	*HOX* Genes with a Significant KM Correlation
Brain-LGG	*HOXA1*, *HOXA4*, *HOXA7*, *HOXA11*
	*HOXB13*
	*HOXC4*
	*HOXD3*, *HOXD4*, *HOXD8*, *HOXD9*, *HOXD10*
Brain-GBM	*HOXB2*, *HOXB9*
Leukemia-LAML	*HOXA7*

^1^ Differentially expressed *HOX* genes for which the KM analysis demonstrates a correlation between increased expression level and patients’ poor survival (*p*-value < 0.05 Bonferroni adjusted for *n* = 39).

**Table 4 cancers-12-01572-t004:** *HOX* gene pairs with significant expression correlation categorized by cancer type ^1^.

Cancer Type\Gene Name	*HOX* Pairs										
	*HOXA5*	*HOXA9*	*HOXB3*	*HOXB5*	*HOXB6*	*HOXB7*	*HOXC8*	*HOXC9*	*HOXC10*	*HOXD9*	Total
**ESCA**											
*HOXA10*					✓	✓					3
*HOXB5*					✓	✓					2
*HOXB6*						✓					1
*HOXC10*							✓				1
**GBM**											
*HOXA3*									✓		1
*HOXA10*									✓		1
*HOXB2*			✓								1
*HOXB4*				✓							1
*HOXB5*					✓						1
**LAML**											
*HOXA3*	✓										1
*HOXA10*		✓									1
**LIHC**											
*HOXD3*										✓	1
*HOXD8*										✓	1
**LUSC**											
*HOXC8*								✓			1
**STAD**											
*HOXC6*							✓	✓			2

^1^ The √ indicates *HOX* gene pair with significant expression correlation.

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
