# Peer review of "Quantitative Analysis of Differential Expression of HOX Genes in Multiple Cancers"

_cancers, 2020, doi:10.3390/cancers12061572_

Round 1

Reviewer 1 Report

The present revised version is now improved with additional experiments.

Some minor corrections are requested :

The explanation for M&M chapter “4.8 KM survival analysis based on expression of HOX gene pairs” (page 19-20should be improved and completed with information that needs to be added in the table/figures legends: the two values N1 and N2 should be given; are the KM plots performed only of the N1 and N2 number of patient samples? The total number of sample should also be given as N to have an overview of the percentage of N1/N and N2/N.

The quality of Table 4 should be improved

Supp table S3 : Please indicate values instead of arrows

Supp table S6 : Please give the p-values for single (lanes 6 and 9) and double gene expression correlation (instead of “1” labels)

The legends of most tables should be improved by adding explanation regarding columns names.

Author Response

Some minor corrections are requested :

The explanation for M&M chapter “4.8 KM survival analysis based on expression of HOX gene pairs” (page 19-20should be improved and completed with information that needs to be added in the table/figures legends: the two values N1 and N2 should be given; are the KM plots performed only of the N1 and N2 number of patient samples? The total number of sample should also be given as N to have an overview of the percentage of N1/N and N2/N.

  • We thank the reviewer the comment. The Materials and Methods section 4.8 and the legend of Figure 6 (now Figure 5) were updated. In addition, a legend has been added to Supplementary file 1.

The quality of Table 4 should be improved

  • Following the comment, the format of table 4 has been revised and improved.

Supp table S3 : Please indicate values instead of arrows

  • The arrows in Supplementary Table S3 (now Supplementary Table S2) were replaced by values of differences in expression levels. Values of up-regulated HOX genes are marked in red and values of down-regulated HOX genes are marked in blue.

Supp table S6 : Please give the p-values for single (lanes 6 and 9) and double gene expression correlation (instead of “1” labels)

  • As suggested by the reviewer, Supplementary Table S6 (now Supplementary Table S5) has been revised. The pair label indication was replaced with the p-value. Additionally, the table structure and format were improved, and a table legend was added.

The legends of most tables should be improved by adding explanation regarding columns names.

  • We thank the reviewer for the comment. The tables were revised and where needed, detailed column legends were added.

Reviewer 2 Report

After revision, the authors have significantly improved the quality of the manuscript. However, I find the representation of the clustering analyses still problematic. I appreciate that the selection of 100 sample to be visualised is  for presentation, but it does provide a biased view of the clustering itself and scores. The authors should choose a better way to visualise their data if the clustering of the whole dataset is not suitable. For instance they could show only the complete dendogram without the heatmap. Dendograms with associated p-values of clustering can be obtained by Bootstrap analysis.

Another small suggestion is to move the present Figure 1 to supplementary.

Author Response

After revision, the authors have significantly improved the quality of the manuscript. However, I find the representation of the clustering analyses still problematic. I appreciate that the selection of 100 sample to be visualised is  for presentation, but it does provide a biased view of the clustering itself and scores. The authors should choose a better way to visualise their data if the clustering of the whole dataset is not suitable. For instance they could show only the complete dendogram without the heatmap. Dendograms with associated p-values of clustering can be obtained by Bootstrap analysis.

  • Following the reviewer’s comment, heatmaps that include all data were created and incorporated into both the manuscript and the supplementary material (replacing the original heatmaps).

Another small suggestion is to move the present Figure 1 to supplementary.

  • Following this suggestion, Figure 1 has been moved to the supplementary material (it is now marked as Supplementary Figure S1).

This manuscript is a resubmission of an earlier submission. The following is a list of the peer review reports and author responses from that submission.

Round 1

Reviewer 1 Report

In the manuscript the authors provide a quantitative analysis of HOX gene expression across multiple cancer types compared to the respective normal tissues. Although the approach is very informative, the study presents important shortcomings that need to be address.

1) The authors assess the validity of the GTEx database as normal tissue control by testing how reproducible are the data compared to the  normal tissues contained in the breast cancer TCGA cohort. This approach should be tested across all other tumour types for which normal tissue is available in the TCGA.

2)  The approach used to discriminate  HOX gene expression between cancer and normal samples is questionable. By randomly comparison of 100 samples at the time, different expression clustering my be obtained. The authors should repeat these analyses using whole tumour cohorts in order to provide a meaningful statistical analysis for differential gene expression.

2) At present, the study only provides an overview of HOX gene expression (upregulated or downregulated) in different tumour types which is not sufficiently novel. By performing a proper clustering analysis, different a and more informative conclusions my be achieved.

3) The manuscript requires restructuring. Discussion should be removed from results and conclusions should be moved after the discussion. Figure 6 is of poor quality, this should be improved.

Reviewer 2 Report

The present manuscript entitled “Pan-cancer quantitative analysis of differential expression of HOX genes” Adato et al focus HOX genes expression and co-expression in several cancer models based on public data (TCGA and GTEx) form comparison with healthy tissues.

The topic of the manuscript is very interesting as there is an increasing interest in HOX genes in cancers since the last few years.

The present manuscript does not aim at compiling the results but at presenting new analyses. However, there is too little comparison with already published data to have a clear overview of the differences/concordances of the present data with that of the literature. This needs to be addressed prior to any re-submission.

The manuscript title is about “Pan-cancer” HOX gene expression analyses but, finally, only a few number of cancer subtypes were selected based on the necessity of a minimum of healthy/cancer samples in each datasets. Consequently, the title should be changed accordingly, as should be a lot of sentences along the manuscript that currently overconcluded. For instance those that argues that “Cancer types … (with) … HOX genes… altered expression… derived from endodermal organs” (as an example from lines 164-167, but multiple other are present along the manuscript up to in the very last part of the conclusion) : since there is a selection of cancer type that have the highest number of sample, it is restricted to sub-types that are well studied or necessitated a huge number of samples for the studies they originate from. Even the last sentence of the conclusion section ended by an uncertain conclusion from the authors: “it can be speculated that”…

Also, as the authors claim that “ By and large, our findings agree with previous reports about the roles specific HOX genes play in cancers” (lanes 290-291), references have to be cited and comparison made in a new table or point by point along the manuscript.

One aim of the present manuscript is to argue that some HOX genes may serve as prognostic markers in different cancers. There are a lot of other prognostic markers validated in the studied cancer models and none of them are indicated as validated in the present study (as KM survival curves but also as (heatmap) differentially expressed genes to validate results on HOX genes. Co-expression with those commonly used markers would be useful too.

The fact that glioblastoma samples express high levels of multiple HOX genes is not that surprising since the expression level of HOX genes is very low (absent) in normal brain tissues: how is the level of expression of those HOX genes relatively to housekeeping genes in the different cancer types evaluated in the manuscript? A graph presenting the mean and dispersion of all samples (healthy vs cancerous) using box-and-whisker with scatter dots plots for a housekeeping gene of reference and the different HOX genes expression would help the interpretation: first, are over-expressed HOX genes really highly expressed?  and second, how is the dispersion and the samples dots ?

For glioblastoma, as cells from brain present roughly no proliferation, a huge number of genes have their expression that is activated in glioblastomas. Therefore, it is not really surprising that HOX genes, together with many other transcription factors or other functional genes associated with cell proliferation/cell cycle/cell death/migration/dedifferentiation status/etc are re-expressed. Please comment on the number of genes that are expressed and how this can impact the comparison of cancerous vs healthy samples.  

It is well known in LAML that the over-expressions of some HOX genes are associated with the prognostic values but are also interesting targets in numerous sub-types of AML. The present results using TCGA-LAML does not reflect the implication of HOXA9/HOXA10 for instance, nor the fact that HOXA9 and HOXA10 (together with other HOXA cluster genes such as HOXA5-6-7) are frequently co-expressed in AMLs.

Similarly, for breast cancer vs normal tissue studies, it is not clear if the authors restricted their analyses to BRCA1-negative cases or not (see Figure 2 header versus text). As it is known that HOX gene expression is different in BRCA1-positive versus negative samples, it is intriguing to focus on “BRCA-“ samples.

Minor points:

There are a lot of repeated sentences along the manuscript. The first two sentences from the abstract and the introduction sections are typical examples: in abstract “HOX genes are a subset of homeotic genes that control the body plan of an embryo along the head-to-tail axis. HOX genes encode transcription factors that function as critical master regulators during embryogenesis” vs in introduction “HOX genes are a subset of homeotic genes that control the body plan of an embryo along the head-to-tail axis. They encode transcription factors that function as critical master regulators during embryogenesis in diverse processes …”. Please read again the manuscript to avoid repetitions that may dilute the messages.

For heatmap figures: how many samples ? restricted to 100 per group? Why not presented the whole datasets?

Figure 1 is unclear to me. The meaning of it should be better explained.

Co-expression with regression curves, R/R2 and p-values should be given too.

Figure S14-15 : the colors of the ribbons of the circus plot are not clear when linking cluster A to B or central to posterior genes for instance.

Supplementary Figure S4 : the rational for “2-groups”, “3-groups” or “quartile” analyses is unclear and should be harmonized or explained : how were the cut-off values determined ? Sometimes, it is indicated that the analysis is a “3-group” one but only two curves are presented (LGG/HOXA2-A6-B13 and so on…)

Conclusion

Despite the fact that the object of the publication could be of real interest for the readers, the manuscript in the present form is to my point of view too preliminary and consequently suffers from some over-conclusion.